# Risk Factors for Severe Outcomes Among Pediatric Cancer Patients with Respiratory Viral Infection

**DOI:** 10.3390/microorganisms13112628

**Published:** 2025-11-19

**Authors:** Alon Kristal, Avi Magid, Nira Arad-Cohen, Moran Szwarcwort-Cohen, Yael Shachor-Meyouhas

**Affiliations:** 1The Ruth & Bruce Rappaport Faculty of Medicine, Technion-Israel Institute of Technology, Haifa 3200003, Israel; kristal.alon@gmail.com; 2Management, Rambam Health Care Campus, Haifa 3109601, Israel; 3School of Public Health, Ben-Gurion University of the Negev, Beer-Sheva 8410501, Israel; 4Department of International Health, Maastricht University, 6200 MD Maastricht, The Netherlands; 5Department of Pediatric Hemato-Oncology, Rambam Health Care Campus, Haifa 3109601, Israel; n_arad-cohen@rambam.health.gov.il; 6Virology Laboratory, Rambam Health Care Campus, Haifa 3109601, Israel; m_szwarcwort@rambam.health.gov.il

**Keywords:** pediatric, hematology-oncology, respiratory viral infection

## Abstract

Viral respiratory infections pose a significant risk for pediatric cancer patients and may lead to a delay in chemotherapy, prolonged hospitalization, and mortality. Limited data exist regarding the contributors to adverse clinical outcomes. The present study aims to describe the associations between clinical, epidemiological, and laboratory factors and severe outcomes of respiratory viral infections among children with cancer. This was a retrospective cohort study among pediatric cancer patients treated in the Pediatric Hematology–Oncology Department at Rambam Health Care Campus from 2016 to 2022. Patients with a positive rt-qPCR test for one of the following viruses were included: Adenovirus, Respiratory Syncytial Virus (RSV), Human Metapneumovirus (HMPV), SARS-CoV-2, Parainfluenza, or Influenza. Demographic, clinical, and laboratory data were collected for each case. GEE analyses were conducted to assess the associations between independent variables and severe outcomes (admission to the Pediatric Intensive Care Unit (PICU), hospitalizations exceeding seven days, co-bacterial infections, and mortality within 30 days). A total of 366 viral infections episodes were identified among 238 patients. There were 187 (51%) children with hematological malignancies, 113 (31%) with solid tumors, and 66 patients (18%) who had undergone bone marrow transplantation. Influenza was the most frequently detected virus, accounting for 89 events (24%), followed closely by Adenovirus, with 82 events (23%). Among the 38 severe events, prolonged hospitalization was the most prevalent outcome, occurring in 33 cases. Adenovirus infection was significantly associated with severe outcomes (OR = 2.97, *p* = 0.010), and antibiotic therapy was associated with 3.62 times higher odds of severe outcomes (*p* = 0.010). Patients presenting with O2 saturation levels below 92% had 5.71 times higher odds of experiencing severe outcomes. Among the subgroup of hematological malignancies, RSV was positively associated with severe outcomes (OR = 4.08, *p* = 0.048). Adenovirus was associated with severe outcomes in pediatric cancer patients, highlighting its prevalence and potential for treatment. Similarly, RSV was associated with adverse outcomes specifically among hematological cancer patients, emphasizing the importance of vaccination. A very low mortality from viral infection was also notable.

## 1. Background

Viral respiratory infections (VRIs) pose a significant risk of morbidity among pediatric patients, contributing to a burden on hospitalization and intensive care unit admission, as well as an economic burden on the health care system [1]. While the majority of these infections tend to exhibit mild clinical courses, their impact on vulnerable groups, such as children with underlying health conditions, requires further research [1]. VRIs particularly impact pediatric hematology–oncology patients, and their presence in this immunosuppressed population is associated with higher mortality, disability, and a delay in chemotherapy, which may indirectly affect survival rates [2]. Previous studies showed that the most frequent viruses among hematology–oncology pediatric patients were Respiratory Syncytial Virus (RSV), Parainfluenza, Influenza type B, Human Metapneumovirus (HMPV), and Human Coronavirus (HCoV) [1,3]. It was also shown that children treated with immunosuppressant agents and children who had undergone bone marrow transplantation (BMT) are more susceptible to VRIs [4,5]. Moreover, concerning the disease course, pediatric hematology–oncology patients may experience a prolonged duration of illness, and are more likely to be treated with antibiotics [1]. Some studies demonstrated the associations between viral infections and prolonged illness in oncology pediatric patients with febrile neutropenia [6,7]. Data on severe acute respiratory syndrome Coronavirus 2 (SARS-CoV-2) among pediatric oncology patients indicate that, although most patients have favorable Coronavirus disease 2019 (COVID-19) outcomes, the mortality is higher compared to children without comorbidities [8,9,10]. Whether or not to delay chemotherapy treatment due to COVID-19 in these patients is questionable. Therefore, the risk of cancer progress or relapse due to a delay in chemotherapy should be carefully considered against the risk of COVID-19 complications and potential mortality [8,9,10]. In a previous study at our institution, we have found that oncological patients with a diagnosed viral infection did not have any difference from those with suspected viral infections and over a one-year study; no complications or mortalities were observed [4]. We aimed to identify specific factors that may be associated with severe outcomes of respiratory viral infection among pediatric oncology patients.

## 2. Methods

### 2.1. Study Design

This was a retrospective study among patients treated in the Pediatric Hematology–Oncology Department at Rambam Health Care Campus from 1 January 2016 until 31 December 2022.

### 2.2. Study Population

The study cohort included patients treated in the Hematology–Oncology Department who were diagnosed with VRIs, confirmed with a positive polymerase chain reaction test.

A swab was drawn from the patient’s throat and both nostrils. The specimens were placed into sterile viral transport media (Virocult, blue cap, MW950; Medical Wire & Equipment Co. [Bath] Ltd., Corsham, Wiltshire, England) and immediately transported to the laboratory or were kept at 4 °C for no longer than 2 days.

Total nucleic acids (NA) were extracted from respiratory samples, using the easymag nuclisense or eMAG instruments (Biomerieux, Marcy l’Etoile, France). Sets of primers and probes were used to detect eight viruses through a multiplex quanlitative real-time reverse-transcription PCR (RT-real-time PCR). Each sample was tested in parallel, in three test tubes, for the following viruses: 1. Influenza A, Influenza B, Respiratory Syncytial Virus (RSV), and Human Metapneumovirus (hMPV); 2. Parainfluenza 1–3; and 3. Adenovirus. Amplification was carried out with the AgPath-IDTM One-Step RT-qPCR Kit (Ambiom, Foster City, CA, USA, Applied Biosystems, Foster City, CA, USA). SARS-CoV-2 was tested using the commercial Seegene kit (Seegene, Seoul, South Korea). The real-time RT-qPCR was performed using the ABI 7500 instrument (applied biosystems) manufactured in Singapore, or the CFX instrument (after March 2020) (Bio Rad, Hercules, CA, USA).

For each patient, every event that was recorded between 2016 and 2022 was included.

Different viral pathogens occurring more than one week apart were considered as separate events. If two infections with the same viral pathogen occurred within one month, a pediatric infectious disease expert assessed the cases to determine if they represented the same event or separate occurrences.

For statistical analysis, each infection event was treated as an independent unit of information instead of aggregating by individual patients, allowing for the gathering of distinct data for each event based on the established criteria.

### 2.3. Inclusion and Exclusion Criteria

The inclusion criteria for infection events were as follows:Patients diagnosed with cancer or those who received BMT visiting the Pediatric Hematology–Oncology Department.Positive polymerase chain reaction (rt-qPCR) tests for viral pathogens.

The exclusion criteria were all the cases which were not included.

### 2.4. Data Collection

For each event, clinical, laboratory, and demographic information was collected. The dependent variables studied included several adverse clinical outcomes assessed as dichotomous variables:Mortality within 30 days.Hospitalization for the infection event exceeding seven days (prolonged hospitalization).Admission to the Pediatric Intensive Care Unit (PICU) due to the infection.Diagnosis of bacterial infection after the viral diagnosis (i.e., bacteremia or pneumonia, where pneumonia is defined as lobar pneumonia, with the presence of bacteria in the blood (i.e., pneumococcus), or a diagnosis of presumed bacterial pneumonia made by an infectious disease expert at the time of diagnosis).

A severe outcome was defined as the occurrence of at least one of the above outcomes.

Independent variables encompassed the following:
Virus type: This was identified via rt-qPCR testing on respiratory specimens, including pathogens such as Adenovirus, RSV, HMPV, SARS-CoV-2 (after March 2020), Parainfluenza, and Influenza types A and B.Demographic data: These included age (categorized into four groups: 0–2 years, 2–5 years, 5–12 years, and >12 years) and sex.Clinical data: Oncological diagnoses were classified into three groups based on the type of cancer and bone marrow transplantation status: hematological cancer, solid tumors, and those undergoing bone marrow transplantation (BMT). BMT status was classified based on a timeline of less than two years from transplantation or more if a diagnosis of graft-versus-host disease was determined. Additional clinical variables included vital signs at diagnosis (fever, oxygen saturation, respiratory rate), weight, therapies administered post-diagnosis (including antibiotics, steroids, IVIg, and antivirals), ongoing medications, and prior respiratory disease diagnoses within one month of the viral diagnosis.Laboratory tests: Laboratory measurements closest to the viral infection event were collected, along with their respective highest or lowest values recorded within three days before and after the event. These included blood counts (WBC, neutrophils, lymphocytes, platelets, hemoglobin) and serum chemistry values (AST, ALT):
○Blood count:
▪White blood cells (WBC): The absolute value was collected, and two new variables were defined depending on the count: leukocytosis for patients with a WBC count higher than 11,000 cells/µL and leukopenia for patients with a WBC count lower than 4000 cells/µL.▪Neutrophils: The absolute count was collected, and the patients were divided into four groups of neutropenia degree depending on the count: severe neutropenia for a neutrophil count lower than 500 cells/µL, moderate neutropenia for 500–1000 cells/µL, mild for 1000–1500 cells/µL, and no neutropenia for patients with a neutrophil count higher than 1500 cells/µL.▪Lymphocytes: The absolute count was collected, and the patients were divided into four groups of lymphopenia degree according to the same values as defined for neutropenia (0–500, 500–1000, 1000–1500, and higher than 1500 cells/µL).▪Platelets: The absolute count was collected, and a new variable for thrombocytopenia was defied as positive if the platelet count was lower than 150,000 cells/µL.▪Hemoglobin: The absolute value was collected, and a new variable for anemia was defined as positive if the hemoglobin count was lower than 11.5 g/dL.
○Chemistry:
▪AST: The absolute level was collected, and the patients were divided into two groups, above and below the threshold of 40 U/L.
○ALT values: The absolute level was collected, and the patients were divided into two groups, above and below the threshold of 65 U/L.


### 2.5. Statistical Analysis

Data were extracted from Rambam’s medical systems using MD-Clone software version 6.3, which facilitates comprehensive access to patient records and allows for flexible data collection based on user-defined criteria. Any additional data was retrieved from the medical records. Descriptive statistics (mean, standard deviation, median, percentiles, and ranges) were calculated using Microsoft Excel and SPSS version 27 for each variable. Normality tests for all the study quantitative variables were conducted using the Shapiro–Wilk test. Whenever needed, parametric and non-parametric tests, including one-way ANOVA, the *t*-test, and the Kruskal–Wallis test, were conducted according to the normality tests’ results. Differences between categorical variables were examined using the Fisher exact test or Pearson χ^2^ test.

Since each infection event was treated as an independent unit of information, in some cases, the same patients were counted a few times. To account for repeated measures, Generalized Estimating Equation (GEE) models were used to assess the associations between the presence of viral infections, clinical variables, and the likelihood of severe outcomes; we conducted six GEE models where, in each model, only one viral pathogen was added as an independent variable to minimize overlap, and the other covariates were the same in each model and included demographic and clinical variables.

Statistical significance was determined using two-sided tests, with *p*-values set at <0.05.

### 2.6. Ethical Considerations

The study is a retrospective study, approved by the local IRB (approval number RMB-D-0003-22).

## 3. Results

During the study period, a total of 1407 respiratory infection events were documented among 666 children treated in the Hematology–Oncology Department. Of these events, 1326 cases underwent rt-qPCR testing for viral pathogens. After excluding events with negative rt-qPCR results, as well as events involving patients treated for non-cancer-related conditions, there were 845 events. Subsequently, repeated cases or cases considered prolonged continuous infections, as determined by a pediatric infectious disease expert, were excluded, resulting in a final inclusion of 366 positive events among 238 children.

### 3.1. Demographic Characteristics

The characteristics of the study population, consisting of 238 children, are summarized in Table 1, while the statistical distribution of the infection events (N = 366) is depicted in Table 2. The cohort included 128 males (54%) and 110 females (46%), with ages ranging from 1 month to 23 years (only 9 patients aged 18 years and above). The median age at presentation was 5.15 years. The total number of events per patient during the study period varied, with 145 patients (61%) experiencing a single event, 63 patients (26%) having two events, 25 patients (11%) with three events, and 5 patients (2%) with four different viral respiratory events.

### 3.2. Viral Pathogens

The most frequently identified viral pathogen was Influenza (24% of total events), followed closely by Adenovirus (23%) and Parainfluenza (19%). RSV was detected in 65 events (18%), while HMPV and SARS-CoV-2 were the least prevalent, each in 31 (8%) and 28 (8%) events, respectively. No virus–virus coinfections were detected. In terms of cancer diagnosis background, 187 events (51%) occurred among individuals with hematological malignancies, 113 events (31%) occurred among patients diagnosed with solid tumors at the time of viral infection, and 66 events (18%) occurred among patients who had undergone bone marrow transplantation (BMT). Patients were classified as BMT recipients only if they had undergone the procedure within two years prior to the viral infection or had graft-versus-host disease necessitating immunosuppressive therapy.

### 3.3. Severe Outcomes

There were 38 severe outcome events (10%). The predominant severe outcome was prolonged hospitalization—exceeding seven days—which accounted for 33 events (9% of total events, representing 87% of severe outcomes). A bacterial infection diagnosed after the viral diagnosis was observed in two cases (0.5% of total events, contributing to 5% of severe outcomes). PICU admissions were also documented in two cases (0.5% of total events, 5% of severe outcomes); one of them was diagnosed with Acute Lymphoid Leukemia, and the other one went through BMT. The only mortality case occurred in a 7-year-old male patient, diagnosed with T-cell Acute Lymphoblastic Leukemia, who had an RSV lower respiratory infection, resulting in subsequent clinical deterioration.

Regarding co-bacterial infections, there were no documented cases of bacteremia related to the viral infection during the study; however, pneumonia was diagnosed in two cases.

Multiple logistic regression analyses were performed for each viral pathogen to evaluate the impact of various factors on the likelihood of experiencing severe outcomes. Notably, a significant correlation was identified specifically for Adenovirus, exhibiting a 2.97-fold increase in the odds of severe outcomes compared to events without Adenovirus detection (*p* = 0.010). Furthermore, the use of antibiotic therapy was associated with a 3.62-fold increase in odds of severe outcomes among children receiving antibiotics during their illness (*p* = 0.010). Patients presenting with O2 saturation levels below 92% upon viral infection diagnosis had 5.71 times higher odds of experiencing severe outcomes (*p* < 0.001).

Other examined variables, including the transplantation status, presence of hematologic malignancy, white blood cell count, sex, and age, did not demonstrate statistically significant associations with severe outcomes (Table 3 and Table 4).

Statistical analysis was conducted for each subgroup of patients based on their type of cancer. It was found that RSV infection was associated with a higher risk of a severe outcome among hematological cancer patients (OR = 4.08, *p* = 0.048) (Table 4). Moreover, the administration of antibiotic therapy following the diagnosis of viral infection was also associated with severe outcomes among hematological cancer patients (OR = 14.41, *p* = 0.004).

## 4. Discussion

The primary objective of this study was to identify the risk factors associated with a severe course of VRIs in children with cancer. Unlike healthy individuals, where VRIs typically present as a self-limiting disease, these infections in hematology–oncology patients may lead to significant complications, including a delay in chemotherapy and prolonged illness [5,6]. Among the various factors examined, an infection with Adenovirus across the study population, RSV infection among hematological cancer patients, the administration of antibiotics, and oxygen saturation levels upon diagnosis were all positively correlated with severe outcomes.

Adenovirus, a non-enveloped double-stranded DNA virus, can present with a wide range of clinical manifestations. It is more prevalent in children under five years old (6.6%) compared to adults (2.0%) [11]. While Adenovirus infections often result in mild illness—affecting the gastrointestinal system, respiratory tract, and conjunctiva—the course of the disease can become much more severe among immunocompromised patients, with reported fatality rates exceeding 50% in untreated cases of pneumonia or disseminated disease [12,13]. In our study, Adenovirus constituted 23% of respiratory viral events, supporting the findings from separate studies where it accounted for 19% in pediatric cancer patients [2] and 13.3% in patients who underwent hematopoietic stem cell transplantation (HSCT) [14].

The correlation between Adenovirus infection and adverse outcomes aligns with previous studies, such as that of Lo et al., which linked Adenoviral infections to increased morbidity and prolonged hospitalizations in pediatric transplant recipients [2]. We also demonstrated, in a recent study among pediatric patients admitted to our PICU with respiratory viral infection, that Adenovirus was linked to higher mortality [15]. In contrast, an extensive multi-center study indicated that Adenovirus infections were rare in a large cohort of cancer patients, with only a 1.1% prevalence and generally favorable prognoses [16]. The contrasting findings highlight the challenge of early diagnosis and the need for timely intervention when Adenovirus infections manifest with lower respiratory symptoms, potentially complicating their identification. Furthermore, appropriate treatment options, especially Cidofovir and Brincidofovir, may play a role, but some also have unwanted effects (mainly cidofovir) due to their potential renal toxicity, mainly in adults [17,18].

Another significant finding was the positive correlation between RSV infection and severe outcomes within the subgroup of hematological cancer patients. Notably, the only mortality case in our cohort involved a patient with ALL who had RSV. RSV, an enveloped single-stranded RNA virus, is particularly prevalent among infants, contributing to a substantial proportion of hospitalizations [19]. Prior studies have indicated that children with cancer often face high rates of RSV infection, with severe disease frequently observed among immunocompromised populations [2,3]. With regard to RSV, the new vaccines are of great promise to infants, pregnant women, immunocompromised patients, and even the elderly [20]. Over 60 countries worldwide approved the use of the RSV pre-vaccine to protect infants against RSV disease [20,21]. Moreover, some studies showed that the RSV vaccine protects healthy infants against hospitalization for RSV-associated lower respiratory tract infection [22], which, together with the findings of our study, emphasizes the importance of RSV vaccination in immunocompromised pediatric patients.

Additionally, we observed a significant association between low oxygen saturation levels measured at the time of diagnosis and the likelihood of severe outcomes. This may express lower respiratory infections with the viruses, which may be of importance in comparison with upper respiratory infections. Prior research also found an association between lower oxygenation levels and prolonged hospitalizations, reinforcing the importance of the continuous monitoring of oxygen saturation, particularly in immunocompromised children [23]. In addition, Kobialka et al. reported a correlation between lower oxygen saturation and a higher risk of PICU transfer or prolonged hospitalizations among hospitalized children [24]. While these studies focused on otherwise healthy patients, the clinical significance of oxygen saturation measurements in immunocompromised children may hold an even greater importance due to their vulnerable health status.

Only two (0.5%) patients were admitted to the PICU because of the respiratory infection, and mortality was exceedingly rare, with only one case occurring within 30 days post-viral infection. The patients admitted to the PICU were infected with RSV and HMPV, emphasizing these viruses’ potential severity.

In relation to SARS-CoV-2, this study did not reveal any significant association between COVID-19 infection and severe outcomes among patients, corroborating findings from a meta-analysis indicating high survival rates in pediatric cancer patients infected with the virus [25]. However, significant disruptions to treatment protocols, including chemotherapy delays, were noted, highlighting the broader implications of COVID-19 on pediatric oncology care, despite the low mortality associated with the virus [8,26]. A recent study from our institution on pediatric ICU patients and viral respiratory illness revealed that, even among pediatric patients admitted to the PICU, SARS-CoV-2 was related with extended ventilation but not to mortality [15]. More studies are needed, as SARS-CoV-2 has become a common pathogen in the population.

The association between antibiotic administration and severe outcomes observed in this study is interesting but not surprising. Among the viral infectious events, 43% of patients received antibiotics post-diagnosis of a viral respiratory infection. Severe neutropenia was present in 21% of these cases. In hematology–oncology patients, guidelines dictate the use of antibiotics in patients with fever and neutropenia even when a viral etiology is confirmed, owing to the inherent risk of bacterial infection after the viral diagnosis. While the existing literature suggests the potential safety of reducing the antibiotic duration for select patients [27], further prospective studies are necessary to clarify whether antibiotic therapy influences the severity of viral infections or is simply a necessary precaution.

### Limitations

This study has some limitations. The primary limitation of this study lies in its retrospective nature, which complicates the ability to thoroughly evaluate confounding variables such as immune suppression, vaccination status, and disease progression. Additionally, retrospectively collected data may hinder establishing causality, particularly in the context of the observed correlation between antibiotic therapy and severe outcomes. Second, this study was a single-center study. Third, some of the prevalent viruses are not routinely examined (Rhinovirus and enterovirus). Fourth, in some of the events, tests were taken only during or after admission. However, some of the laboratory parameters are not significantly affected by the time the virus is detected. Finally, the years 2020–2023, which are included in the study period, represent the COVID-19 pandemic period. During that period, there were different exposures and isolation procedures that may have affected some of the results.

## 5. Conclusions

In conclusion, this study identifies Adenovirus and RSV infections, as well as low saturation, as significant predictors of severe illness in pediatric cancer patients, but with low PICU admissions and mortality. The prevalence of these viruses with the availability of relevant treatment emphasizes the need for early diagnosis and individualized interventions, and the need for vaccination in order to prevent severe outcomes. Our findings emphasize the importance of RSV prevention among immunocompromised pediatric patients. The correlation between antibiotic administration and severe outcomes should be further investigated.

## Figures and Tables

**Table 1 microorganisms-13-02628-t001:** Distribution of events by selected characteristics (N = 366).

Variables		N	%
Age (years)	Mean ± SD (range)	6.99 ± 5.24 (0.07–23)	
Age group (years)	0–2	63	17
	2–5	110	30
	5–12	115	32
	12+	78	21
Virus type	Influenza	89	24
	Adenovirus	82	23
	Parainfluenza	71	19
	RSV	65	18
	HMPV	31	8
	SARS-CoV-2	28	8
Cancer category	Solid tumor	113	31
	Hematologic malignancy	187	51
	BMT	66	18
Severe events	Mortality within 30 days	1	0.3
	Bacterial infection after the viral diagnosis	2	0.5
	PICU hospitalization	2	0.5
	Prolonged hospitalization (>7 d)	33	9
WBC (×10^3^)	Mean ± SD (range)	5.12 ± 4.8 (0.03–28.9)	
	Median (IQR)		
Neutrophils	Mean + SD	3.42 ± 3.3	
Neutropenia category	No neutropenia	228	62
	Mild	28	8
	Moderate	34	9
	Severe	76	21
Leukocytosis	Yes	44	12
	No	322	88
Lymphocytes	Mean + SD	1.80 ± 1.7	
Anemia	Yes	253	69
	No	113	31
Antibiotic therapy after	Yes	159	43
diagnosis with viral infection	No	207	57
O_2_ Saturation at the time	≥92%	328	92
of diagnosis with viral infection	<92%	38	8

**Table 2 microorganisms-13-02628-t002:** GEE model: risk factors for severe events (dependent variable).

Variables		OR	95% CI	*p*
Adenovirus	No	1		
	Yes	3.15	1.15–7.12	0.010
Sex	Male	1		
	Female	1.55	0.74–5.56	0.235
Antibiotics therapy after	No	1		
diagnosis of viral infection	Yes	3.45	1.40–8.85	0.020
O_2_ Saturation at the time of	≥92%	1		
diagnosis	<92%	5.79	3.15–11.93	<0.001

**Table 3 microorganisms-13-02628-t003:** Correlation between virus type and severe outcome (each correlation represents a different GEE model, where in each model the dependent variable was severe events; all other covariates in each GEE model (except the virus) were the same).

Variables	OR	95% CI	*p*
Adenovirus	3.15	1.15–7.12	0.010
SARS-CoV-2	0.63	0.11–3.54	0.596
Influenza	0.38	0.11–1.34	0.132
HMPV	1.52	0.39–5.95	0.547
RSV	0.97	0.35–2.70	0.953
Parainfluenza	0.64	0.24–1.69	0.362

**Table 4 microorganisms-13-02628-t004:** GEE model: risk factors for severe events (dependent variable) among hematology–oncology cancer patients.

Variables		OR	95% CI	*p*
RSV	No	1		
	Yes	5.02	1.68–10.06	0.032
Sex	Male	1		
	Female	2.25	0.69–7.15	0.281
Antibiotics therapy after	No	1		
diagnosis with viral infection	Yes	12.15	5.55–60.03	0.003

## Data Availability

The datasets presented in this article are not readily available due to privacy issues.

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
