# Peer review of "Risk Factors for Severe Outcomes Among Pediatric Cancer Patients with Respiratory Viral Infection"

_microorganisms, 2025, doi:10.3390/microorganisms13112628_

Round 1

Reviewer 1 Report

Comments and Suggestions for Authors

This is a single-center, retrospective study of respiratory viral infections among pediatric cancer patients. Although the data in this population are quite limited, many things need to be refined before considering publication. Please see details on major and minor issues.

Major issues:

  1. Overall, the manuscript is too long, and I would recommend focusing on the key information in the introduction, results, and discussion.
  2. Some of the recent papers do not seem to be included. Especially, a number of COVID-19 papers have been published in this population
  3. Regarding “For statistical analysis, each infection event was treated as an independent unit of information instead of aggregating by individual patient, allowing for the gathering of distinct data for each event based on the established criteria.”, this means the same patients are counted many times. This is concerning from a statistical point of view, unless an appropriate model (e.g., GEE) is used. I would recommend including biostatisticians as coauthors for appropriate analyses. Otherwise, interpretation can mislead.
  4. Unless the previous literature uses, it does not make sense that combining Mortality within 30 days, Hospitalization for the infection event exceeding seven days (prolonged hospitalization), Admission to the Pediatric Intensive Care Unit (PICU) due to the infection, and Co-bacterial infections (i.e bacteremia or pneumonia) as Severe outcome. These are quite different from an infectious disease and outcome perspective. In fact, the majority of severe outcomes come from prolonged hospitalization.
  5. Regarding tables 2 and 4, it does not make sense to include the variables that occurred after the respiratory viral infection diagnosis. It is unclear which one is the predictor and which one is the result. For instance, if patients had bacterial co-infection, the physicians used antibiotics. However, here, bacterial co-infection was counted as outcome. This concept is also related to the last sentence in the discussion.

Minor issues:

  1. I am not aware of any study counting patients aged up to 23 years old as children.
  2. Regarding “If two infections with the same viral pathogen occurred within one month, a pediatric infectious disease expert assessed the cases to determine if they represent the same event or separate occurrences.”, please provide the data or references indicating the definition are reasonable. Many children, particularly cancer patients, shed respiratory virus for a very long time. If the same virus is found more than one month apart, the second detection is more likely just long-term shedding rather than a new infection.
  3. Exclusion criteria are the mirror of inclusion criteria. Thus, there is no point to mention it.
  4. Regarding “Laboratory measurements closest to the viral infection event were collected, along with their respective highest or lowest values recorded within three days before and after the event.”, it is unclear why we can include values after the event.
  5. Regarding neutropenia and lymphopenia in this context, I would recommend what kinds of cutoff have been used in this context. The values are typically lower than what the authors used. It is unclear of using leukocytosis in this setting.
  6. It was unclear regarding “six multiple logistic regression models were constructed”. An explanation is needed regarding the process of sequence, such as how factors are selected, etc.
  7. It is unusual to show all of mean, SD, range, median and IQR.
  8. The definition of pneumonia are needed. A respiratory virus infection can also be diagnosed with viral pneumonia.
  9. Based on table 3, it is unclear which one is the reference group.
  10. The definition of co-bacterial infection are a little bit unclear.
  11. Regarding Table 4, I assume the number of severe events is much less than 38. In such a case, how could the authors include at least 4 variables into one multivariable model?
  12. Regarding, “Furthermore, appropriate treatment options, especially with agents like Cidofovir and Brincidofovir, may play a role but some also have unwanted effects due to potential renal toxicity, mainly in adults (18, 19).”, I would disagree. Cidofovir can cause significant renal toxicity in children and adults. Bricidofovir is much less likely to cause renal toxicity.

Author Response

Dear reviewer,

Many thanks for reviewing our manuscript. We thoroughly read your review and corrected the answers accordingly. Our answers to each of your comments can be found in the attached file.

Reviewer 2 Report

Comments and Suggestions for Authors

1) introduction section is just one parragraph, this make difficult to read. Plese improve the quality of the introduction section, and add more information form other articles in cancer patientes. There is more information that the authors employed in the introductions section.

2) Real time reverse-transcription PCR is not RT-PCR, is RT-qPCR. Please correct it.

3) In the statistical section please mentioned which test was performed to determine data distribution

4) It's not correct that the authors reports mean and median of age and WBC. It data follows normal distribution reports mean and SD, if data did not follow normal distribution please reports mediand and IQR.

5)  In table 1 not only report WBC as a continuos variable, also neutrophils and lympocytes. 

6) Table 2, 3 and 4 is not clear which is the dependet factor  evaluted with the regresion (is not clear "severe event". Also is not clear if there are several binomial logistic regression or a multiple logistic regresion. Please, first perfome a binomial logistic regression and then a multiple logistic regresion. 

7) If authors performed regressions to calcualte OR of severe event, in table 1, add Sever patients and non-severe patientes, and compare

8) also evalute duration of hospitalization.

Author Response

(The authors gave the same response as above.)

Reviewer 3 Report

Comments and Suggestions for Authors

The submitted manuscript concerns the links between clinical, epidemiological, and laboratory factors and the serious consequences of respiratory viral infections in children with cancer. The topic addressed by the authors is interesting, although I have a few comments.

  1. The abstract should be a total of about 200 words maximum. Please shorten the abstract.
  2. Reference numbers should be in square brackets. Please correct these.
  3. Line 60: Names used for the first time should be written in full. Please expand the term RSV.
  4. Line 80: As above. Please expand the term SARS-CoV-2.
  5. Lines 80-81: It is not true that data on SARS-CoV-2 in pediatric oncology patients are sporadic. Below are links to several publications, and there are quite a few more in the PubMed database. Please find the relevant literature and revise the manuscript to include the additional data.

DOI: 10.1007/s00431-021-04338-y

DOI: 10.1093/infdis/jiad496

DOI: 10.7759/cureus.46149

DOI: 10.1186/s13045-021-01181-4

  1. Lines 111-112: The authors introduced full names and abbreviations in the Introduction. Please use abbreviations here.
  2. Line 115: Please add the country of manufacture for the ABI 7500 instrument (Applied Biosystems).
  3. Line 218: Are patients over 18 years of age considered pediatric patients? Please explain why these 9 patients were included in the study.
  4. Line 219 and Table 2: Do the authors have any literature data indicating that Jews and/or Christian Arabs manifest viral diseases differently? If not, please remove such data divisions from the entire manuscript.
  5. Did the authors detect virus-virus coinfections? If not, please include this in section 3.2. Viral Pathogens.
  6. Line 243: Please use the abbreviation BMT, which the authors introduced in line 233.
Comments on the Quality of English Language

The manuscript requires professional linguistic editing.

Author Response

(The authors gave the same response as above.)

Reviewer 4 Report

Comments and Suggestions for Authors

Kristal et al. have reported on risk factors of poor clinical outcomes among pediatric cancer patients with respiratory virus infections. The work is interesting; the short, straightforward presentation is well in line with retrospective study-design and the small sample count. The discussion of the results occurred in a balanced way, study limitations are admitted appropriately.

I have a few recommendations on how the manuscript can be further improved.

  1. Methods: For stylistic reasons, I recommend replacing the bullet points by a table.
  2. Methods/results: It should be better explained why the authors specifically addressed 92% O2 saturation as the cut-off for their statistical assessment. Would the results have been different if, for example, 91% or 93% would have been chosen as a cut-off instead? The authors may at least want to comment on this point in their discussion.
  3. Discussion, lines 329-338: It is a pity that the COVID-19-related findings were only shortly discussed. It is interesting that increased proportions of severe outcomes were not observed even in spite of “significant disruptions to treatment protocols” as reported by the authors. Have these “significant disruptions to treatment protocols” been due to a poor medical condition of the patients associated with the virus infections or due to organizational difficulties associated with enforced anti-SARS-CoV-2-countermeasures? If yes, could the problem be solved when these countermeasures were eased / partly lifted at later stages of the pandemic? Your respective experience could be highly valuable for the management of future similar pandemic situations.

4. Discussion, lines 339-347: Can you differentiate between antibiotic treatment just applied due to guideline requirements and antibiotic treatment due to suspected or proven systemic bacterial infections? If yes, a sub-group analysis would be desirable to discriminate effects of poor general condition and effects of true bacterial infections.

Author Response

(The authors gave the same response as above.)

Round 2

Reviewer 2 Report

Comments and Suggestions for Authors

The manuscript has been improved enought to be published, despite the low soundess

Author Response

Dear reviewer,

Many thanks for your report. Since you mentioned that the manuscript had been improved enough to be published, we would like to thank you again for reviewing our manuscript.

Reviewer 3 Report

Comments and Suggestions for Authors

I am not satisfied with the argument that clinical and epidemiological studies conducted in Israel distinguish between Jews and Arabs. The cited publication states that ethnic minorities (tribes) that are poor, live in overcrowded conditions, and do not have access to adequate medical care may be more susceptible to infection.
The authors do not indicate in their work that Arab patients lived in poverty and overcrowding. Since the authors included them in their research, it means that they had access to medical care.
All of this confirms my belief that the division made by the authors is artificial and based solely on ethnicity. Furthermore, given the current geopolitical situation in the region from which the results presented originate, I believe that divisions among the population living in this area should not be reinforced.

Author Response

Dear reviewer,

Many thanks for reviewing our manuscript. Here is your comment after the second review, followed by our reply.

Reviewer comment: I am not satisfied with the argument that clinical and epidemiological studies conducted in Israel distinguish between Jews and Arabs. The cited publication states that ethnic minorities (tribes) that are poor, live in overcrowded conditions, and do not have access to adequate medical care may be more susceptible to infection.
The authors do not indicate in their work that Arab patients lived in poverty and overcrowding. Since the authors included them in their research, it means that they had access to medical care.
All of this confirms my belief that the division made by the authors is artificial and based solely on ethnicity. Furthermore, given the current geopolitical situation in the region from which the results presented originate, I believe that divisions among the population living in this area should not be reinforced.

Our reply:

We totally agree with you. After considering it again, we agree that there is no scientific need in this case to divide by ethnicity. We totally agree that divisions among the population living in our area should not be reinforced. Thank you for clarifying it. We appreciate it very much.

We removed the ethnic division from the entire manuscript, including from our analyses (it was not statistically significand in any case, but we repeated the analyses in any case).

Many thanks again for your important review!